# Vitamin D deficiency or supplementation and the risk of human herpesvirus infections or reactivation: a systematic review protocol

Liang-Yu Lin , Ketaki Bhate, Harriet Forbes, Liam Smeeth, Charlotte Warren-Gash, Sinéad Langan

Department of Non-communicable Disease Epidemiology, London School of Hygiene and Tropical Medicine Faculty of Epidemiology and Population Health, London, UK

**Correspondence to**
Prof Sinéad Langan;
sinead.langan@lshtm.ac.uk

## ABSTRACT

**Introduction** Human herpesviruses induce lifelong latent infections and may reactivate as the immune system deteriorates. Recent studies have suggested that vitamin D, an essential element of bone health, may have some effect of protecting against infections, but investigations of its potential to prevent herpesvirus infection or reactivation are limited. We will review the current literature examining vitamin D and the risk of herpesvirus infections or reactivation.

**Methods and analysis** Our systematic review will address two research questions: (1) Do deficient/insufficient serum vitamin D levels increase the risk of herpesvirus infections and (2) Does vitamin D supplementation protect against herpesvirus infections? We will include only intervention studies with control groups, cohort studies and case-control studies. We will use subject headings and keywords to search for synonyms of 'vitamin D' and 'herpesviruses' (including herpes simplex virus type 1 and 2, varicella-zoster virus, cytomegalovirus, Epstein-Barr virus and human herpesviruses type 6, 7 and 8) in Medline, Embase, Global Health, Web of Science, Scopus and Cochrane Central Register of Controlled Trials, and the grey literature databases Open Grey, EThOS and BASE from inception to 31 August 2019. References to the included articles and relevant systematic reviews will also be examined. Two reviewers will independently screen the study titles and abstracts, and examine the full texts to decide the final eligibility. They will independently extract data from the studies and assess bias using the Cochrane Collaboration approach. A third researcher will solve any discrepancies. The results will be narratively synthesised; if an adequate number of studies is included and the homogeneity between studies is acceptable, a meta-analysis will be performed. We will assess the quality of evidence using the Grading of Recommendations, Assessment, Development and Evaluation framework, and display the results in a summary of findings table.

**Ethics and dissemination** Ethical review is not required for a systematic review. We will publish the results in a peer-review journal. Any amendments to the protocol will be recorded in the supplementary section.

**PROSPERO registration number** CRD42019130153.

### Strengths and limitation of this study

► This systematic review will be undertaken following the predefined population, exposure, comparator and outcome framework.
► All available databases, including six major databases and three grey literature databases, will be searched to obtain all eligible studies.
► The summarised results will improve our understanding of the current evidence of the possible association between vitamin D and herpesvirus infection/reactivation.
► The number of sufficient eligible studies may be inadequate, especially for some viruses that are harder to diagnose; also, the included studies may not have an adequate quality of evidence to answer the research questions.

## INTRODUCTION
### Rationale

Herpesviruses are a group of double-stranded DNA viruses that infect humans and some animals. After infecting their hosts, these viruses cannot be eradicated; instead, they establish latency and persist for life. As the host's immunity declines, these viruses can reactivate to induce various symptoms. There are eight human herpesviruses (table 1).[1] Reactivation of herpesviruses may induce serious complications. For instance, herpes simplex virus 1 (HSV-1) can lead to herpetic keratitis, which is the major cause of blindness in high-income countries[2]; Epstein-Barr virus may induce nasopharyngeal cancer[3] and varicella-zoster virus would cause herpes zoster and postherpetic neuralgia, which increases financial burdens, especially for people older than aged 65 years.[4] Consequently, investigating immunomodulatory factors associated with infection or reactivation of this virus family is important.

**Table 1** List of human herpesviruses

| Common name | Abbreviation |
| --- | --- |
| Herpes simplex virus type 1 | HSV-1 |
| Herpes simplex virus type 2 | HSV-2 |
| Varicella-zoster virus | VZV |
| Epstein-Barr virus | EBV |
| Cytomegalovirus | CMV |
| HHV-6 variant A | HHV-6A |
| HHV-6 variant B | HHV-6B |
| HHV-7 | HHV-7 |
| Kaposi's sarcoma-associated HV | KSHV |

Vitamin D is mainly endogenously synthesised by the skin after sun exposure and can be supplied through dietary intake and supplementation. It plays an important role in absorbing calcium and phosphate, which are essential for bone health.[5] Recently, some studies have indicated that vitamin D may have potential immunomodulatory effects associated with the regulation of antimicrobial peptides (AMPs).[6] In previous cell studies, vitamin D induced gene expression of an AMP named cathelicidin. In response to pathogen exposure, immune cells such as monocytes or macrophages, upregulate vitamin D receptors and enzymes to increase the production of cathelicidin.[7–9] In addition, evidence suggests that vitamin D has some effects on the adaptive immune system. Vitamin D suppresses CD4+ T helper (Th)1 lymphocytes and increases Th2 lymphocytes, and it also intensifies the effect of regulatory T lymphocyte responses.[10 11] Regarding the effects of vitamin D-associated AMPs on herpesviruses, a cell study indicated that cathelicidin decreased HSV-1 viral titres isolated from patients with keratoconjunctivitis[12]; furthermore, another cell study also showed that vitamin D supplementation reduced HSV-1 viral load and mRNA expression in HSV-1-infected cells.[13]

Vitamin D also shows some anti-infective potential in epidemiological studies. A meta-analysis using original patient data from 25 randomised controlled trials showed that among the general population, vitamin D supplementation reduced the risk of acute respiratory infections.[14] Furthermore, there is some evidence to suggest an anti-infective effect of vitamin D in specific patient groups, such as patients with chronic kidney disease (CKD), human immunodeficiency virus (HIV) or hepatitis C virus (HCV) infection. Among patients with CKD receiving dialysis, a case-control study indicated that the risk of herpes zoster reactivation was significantly lower in those who received vitamin D supplementation[15]; another meta-analysis also showed that patients with CKD with higher or normal serum vitamin D levels had a lower risk of infection.[16] Among HIV-infected patients, lower serum vitamin D levels were also associated with a higher risk of clinical progression to AIDS and all-cause mortality in a cohort study,[17] while vitamin D supplementation

did not affect mortality, CD4 cell count or viral load.[18] For HCV-infected patients, serum vitamin D levels were inversely associated with the grade of liver inflammation and the stage of fibrosis,[19] while no protective effect of vitamin D supplementation was seen in a meta-analysis of clinical trials.[20] However, the effect of vitamin D on herpesvirus infection or reactivation in the general population is unclear.

In this study, we will comprehensively review studies of the effect of serum vitamin D levels or the use of oral vitamin D supplementation on infection with or reactivation of any of the eight human herpesviruses.

## OBJECTIVE
This review aims to explore the association between vitamin D and herpesviruses. The proposed systematic review will address two primary research questions:
1. Is serum vitamin D deficiency/insufficiency associated with an increased risk of infection with or reactivation of human herpesviruses?
2. Does oral vitamin D supplementation protect against infection with or reactivation of human herpesviruses?

## METHODS
This study protocol will be reported according to the Preferred Reporting Items for Systematic Review and Meta-Analysis Protocols.[21]

### Patient and public involvement
Patients and/or the public were not involved in this systematic review protocol.

### Eligibility criteria
#### Study design and characteristics
To identify the possible causal association between vitamin D and herpesvirus infections, we will review observational studies, including cohort and case-control studies, and intervention studies with any type of control group, either placebo, active comparator or no treatment, including randomised or non-randomised controlled trials. Descriptive studies, ecological studies, cross-sectional studies, case reports and case series will not be included. If a systematic review relevant to our topic is found, we will review its references.

#### Participants
Only human studies will be included in our review, and animal or cell studies will be excluded. Studies from all age groups or involving patients with any immune status are eligible.

#### Exposure
We have two exposures of interest for each research question in our review. For the first research question, the exposures are deficient or insufficient serum vitamin D levels in the participants. We will include articles in which the serum vitamin D levels are identified as deficient or

**Table 2** The normal range of serum vitamin D levels adapted in different studies

| Study | Deficiency (nmol/L) | Insufficiency (nmol/L) | Adequate (nmol/L) |
|---|---|---|---|
| Pearce and Cheetham[26] | <25 | 25–50 | 50–75 |
| Institute of Medicine[27] | <30 | 30–50 | ≥50 |
| Holick et al[28] | <50 | 52.5–72.5 | |
| Hanley et al[29] | <25 | 25–75 | |

1 nmol/L=0.4 ng/mL.

insufficient. Notably, no consensus exists about serum vitamin D levels, and different studies may use different definitions (table 2). If eligible studies provide original values for the serum 25(OH)D levels, we will define vitamin D deficiency as 25(OH)D<25 nmol/L and insufficiency as 25(OH)D in the range from 25 to 50 nmol/L in accordance with the standards of Public Health England.[22]

For the second research question, the intervention is oral vitamin D supplementation or oral vitamin D analogue treatment used for secondary hyperparathyroidism or osteoporosis (online supplementary table 1). We will exclude studies using topical vitamin D analogues, because their effects on systemic serum vitamin D levels are unknown.

## Comparators

The comparator groups for vitamin D insufficiency and deficiency are those with sufficient serum vitamin D levels. For those receiving oral vitamin D supplementation or treatment, the comparators are those without vitamin D supplementation or treatment or receiving placebo or another active comparator, respectively.

## Outcomes

The outcomes will be infection with or reactivation of all eight human herpesviruses listed in table 1. Infection with and reactivation of herpesviruses are defined using clinical or laboratory criteria. The clinical criteria include patients presenting with classical symptoms, for instance, the painful rash of herpes zoster, or a diagnosis recorded by physicians. Laboratory criteria include using laboratory techniques to confirm the diagnosis, such as evaluation of the serum viral load by PCR. Studies reporting only serum antibodies against herpesviruses will not be included.

## Information sources

We will search the following database from inception to 31 August 2019: Medline (Ovid), EMBASE (Ovid), Web of Science, Scopus, Cochrane Library and Global Health (Ovid). To enhance the sensitivity of our search and reduce the risk of publication bias, we will also search grey literature databases such as Open Grey, BASE, EThOS and the clinical trials register at ClinicalTrials.gov.

## Search strategy

We will search for synonyms of 'human herpesviruses' and 'vitamin D' using both controlled vocabularies and keywords in each database. A search strategy in Medline (Ovid) is listed in online supplementary table 2. The subject headings will be modified for different databases. The results will be combined using the Boolean logic operator 'AND'. For some database with limited search functions, such as single line search, keywords will be split into small sections to fulfil the requirement (online supplementary table 3). In addition to searching electronic databases, we will manually search for the reference lists of the included articles and relevant systematic reviews.

## Study records

### Data management

One researcher will import the search results from different databases into the citation management software EndNote X9 (Clarivate Analytics, V.9.1/2019). Duplicated results will be identified and deleted.

### Selection process

Two researchers will independently screen the titles and abstracts of all identified studies. We will obtain full texts of all studies fulfilling the review criteria, and the two researchers will screen all articles to establish their eligibility for inclusion. Any discrepancy in the reviewing process will be adjudicated by the third researcher.

### Data collection process

Data extraction for the first three studies will be performed by two independent researchers to ensure the integrity of the process; then, one researcher will extract data from the remaining studies. If any data are missing or unclear, we will contact the authors for further clarification and information.

### Data items

We will summarise and extract data from the included studies using a Population, Exposure/Intervention, Comparator, Outcomes and Study characteristics framework as follows:
1. Population: sample sizes of each study, inclusion or exclusion criteria for the participants and demographic characteristics, such as sex, age or immune status

2. Exposure 1: insufficient or deficit serum vitamin D levels of the study participants. Exposure 2: oral vitamin D supplementation or vitamin D analogue
3. Comparator 1: sufficient serum vitamin D levels among the participants. Comparator 2: without vitamin D supplementation or vitamin D analogue use.
4. Outcomes: definition of herpesvirus infection/reactivation, that is, clinically diagnosed or laboratory-confirmed herpesvirus infections; the number of study subjects with the outcomes
5. Study characteristics: publication details (authors, publication year, country and journal), study designs, confounders measured, confounders adjusted and study results

### Outcomes and prioritisation

The outcomes of the studies are herpesvirus infections or reactivation. The disease can be diagnosed either clinically or through laboratory confirmation. Some herpesvirus infections, such as herpes zoster, may lead to characteristic clinical symptoms, and some laboratory approaches, such as PCR, can detect the viral load, which can also be proof of herpesvirus infection/reactivation. Studies measuring serum antiherpesvirus antibodies will not be included, because antibodies detected in the absence of clinical symptoms cannot ascertain the timing of infection. Regarding the study results, our focus is on the incidence of herpesviruses infection or reactivation. We will report the outcome definition used for each study, and will extract data on the appropriate effect measure. The effect measures will include ORs for case-control studies and risk, rate or HRs for cohort studies or clinical trials. If studies report only continuous outcomes such as viral load, then we will summarise these using means or medians as appropriate.

### Risk of bias in individual studies

We will assess the risk of bias from the included studies using a template form based on the Cochrane approach for trials and observational studies, and all relevant domains of bias will be assessed for different study types. For randomised controlled trials, we will evaluate bias due to the following sources: (1) random sequence generation; (2) allocation concealment; (3) blinding of participants, personnel or assessment; (4) incomplete outcome data and (5) selective reporting. For observational studies, we will consider bias due to the following sources: (1) confounding factors; (2) selection of participants or controls; (3) differential and non-differential misclassification of the exposure or outcome; (4) reverse causation and (5) missing data (online supplementary table 4). A piloted form will be tested by applying it to extract data from the first three studies. To ensure the quality and consistency of the risk of bias assessment, the first three studies will be evaluated by two independent researchers. Then, one researcher will complete the evaluation of the remaining included studies. Any discrepancies will be examined by the third researcher.

### Data synthesis and meta-bias (es)

We will comprehensively search different databases including grey literature databases to minimise bias in our search. We will use a narrative synthesis to summarise the data and results from the eligible studies included in our review. Since each herpesvirus subtypes have different pathogenic pathways, we will display the results separately by virus. If an adequate number of studies with acceptable homogeneity are included, a meta-analysis will be performed to integrate the study results. We will decide whether to use fixed-effects or random-effects models by considering the $I^2$ value for heterogeneity; values >50% will be considered to represent substantial heterogeneity.[23] If the number of included studies is sufficient, we will carry out subgroup analyses of subjects with baseline vitamin D status, intervention trials, study durations, different ages, immune statuses or latitude to explore sources of heterogeneity. A funnel plot will be used to present the distribution of studies and examine any possible publication bias.[24] All statistical analyses will be performed by using STATA V.15.

### Confidence in cumulative evidence

We will evaluate the quality of evidence using the Grading of Recommendations, Assessment, Development and Evaluations framework.[25] We will evaluate the design and risk of bias across all included studies. Studies with significant effects, strong dose responses or that are well adjusted for plausible confounders will receive higher grades, whereas those with a higher risk of bias, inconsistency, indirectness or imprecision will be downgraded. We will conclude the quality of evidence using a 'Summary of Findings' table and assign each outcome a quality rank of 'high', 'moderate', 'low' or 'very low' for the level of confidence.

**Contributors** L-YL contributed to the design of the study, drafted the introduction, methods and analysis and revised the protocol according to other authors' comments; CW-G contributed to the design of the study, made critical comments on the protocol and revised the paper critically; SML contributed to the design of the study, made critical comments on the protocol and revised the paper critically; KB, HF and LS contributed to the design of the study and revised the paper critically. All authors approved the final version of the protocol.

**Funding** LYL is funded by the scholarship of government sponsorship for overseas study by the Ministry of Education Republic of China (Taiwan). CWG is supported by a Wellcome Intermediate Clinical Fellowship (201440_Z_16_Z). SML is funded by a Wellcome Senior Clinical Fellowship in Science (205039/Z/16/Z). KB is supported by a National Institute of Health Research (NIHR) Doctoral Research Fellowship (DRF-2018-11-ST2-066).

**Competing interests** None declared.

**Patient consent for publication** Not required.

**Ethics approval** Ethical review is not required for a systematic review. We will publish the results in a peer-review journal. Any changes in the study protocol will be recorded and reported. Any revision of the protocol will be recorded in the supplementation of the final published review.

**Provenance and peer review** Not commissioned; externally peer reviewed.

**ORCID iD**
Liang-Yu Lin http://orcid.org/0000-0003-4720-6738

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
