## [Reviewer comments · BMJ Open]

ARTICLE DETAILS

TITLE (PROVISIONAL)	Vitamin D deficiency or supplementation and the risk of human herpesvirus infections or reactivation: a systematic review protocol
AUTHORS	Lin, Liang-Yu; Bhate, Ketaki; Forbes, Harriet; Smeeth, Liam; Warren-Gash, Charlotte; Langan, Sinead

VERSION 1 – REVIEW

REVIEWER	Genevieve Mailhot University of Montreal Canada
REVIEW RETURNED	18-Jun-2019

GENERAL COMMENTS	This systematic review protocol will bridge a gap in our knowledge on vitamin D and herpes virus infections by providing answers to the following questions: 1) Are deficient or insufficient vitamin D levels associated with the risk of herpes virus infections/reactivation? and; 2) Does vitamin D supplementation protect against herpes virus infections/reactivation? Overall, the protocol is clearly written and the approach looks sound. I only request a few clarifications: 1. Rationale: The rationale for undertaking this systematic review is thin. Biological plausibility is only briefly touched upon through the general description of the role of vitamin D in innate immune responses. Given that viral infections involve adaptive immune responses as well, I would have expected the authors to summarize the state of knowledge on the role of vitamin D in the adaptive immune system. The authors cited only two systematic reviews/meta-analyses, which have investigated the role of vitamin D in infections, including a work in individuals with chronic kidney diseases that has addressed infections from a rather broad perspective. In my opinion, the rationale should include a more thorough review of previous observational and interventional studies conducted on vitamin D and viral infections, such as hepatitis C, HIV, and others. 2. Outcomes: Is the focus only on the incidence of infection and reactivation or will you consider reporting continuous outcomes such as viral load, clinical symptoms or other measures? This should be made clearer in the protocol. How will you report outcomes, e.g. as risk or odds ratios, as means or mean differences for continuous measures?
---

	3. Analysis: The authors stated that they will analyze the results separately by virus, given their distinct pathogenic mechanisms. If the number of studies is found insufficient to perform such distinct analyses, what will be the alternate strategy? If the number of included studies is sufficient, please consider including in subgroup analyses the effect of baseline vitamin D status and for intervention trials, study duration.
--	---

REVIEWER	Peter Bergman Department of Laboratory Medicine, Karolinska Institutet, Stockholm, SWEDEN
REVIEW RETURNED	24-Jun-2019

GENERAL COMMENTS	Review of "Vitamin D and human herpesviruses infections: a systematic review protocol" by Lin et al. This protocol describes the plan to perform a systematic review and meta-analysis of the topic of vitamin D and human herpesviruses. The topic is timely, interesting and relevant. Also, the idea to publish the protocol separately is valid. The manuscript is well written and easy to read and understand. However, there are a few minor points that need to be addressed. Title: The title reflects the content in a good way. Abstract: I suggest that the different forms of herpes-virus infections are mentioned in the abstract. Also, it should be mentioned whether placebo-controlled trials will be included or not. This can be important given the inherent problems with bias for all other interventional trials (open, non-controlled, risk of selection bias etc) Strengths and limitations: the concept of grey literature databases needs to be explained somewhere. Rationale: "increased financial burden" needs to be better explained, for whom? Reference 6 is not adequate, also it should be written "antimicrobial peptides, AMPs" to follow the standard nomenclature in the field. You could refer to Zasloff, Nature 2002 for an overview of AMPs or similar overviews from a later date. The connection between Vitamin D and AMPs was first described in 2004 by Wang followed in 2005 by Gombart and Weber in three papers published almost at the same time. These papers should be cited. Rationale section: please provide some more information why vitamin D would prevent herpes virus reactivation? Are there any mechanistic papers supporting such a claim? Study design: please comment on the role of placebo-controlled studies or non-controlled studies. Participants: perhaps a subgroup-analysis should be done for immunocompromised patients? Comparators: please comment on placebo or not Outcomes: this is a tricky section since the symptoms of primary infection may vary across the different herpes-viruses. In addition, symptoms of reactivation can also differ between for example HHV6 and HHV8 and HSV-1. Please, comment how you aim to solve this problem in the analysis. My guess is that you will find a majority of studies for a few viruses (HSV1 and 2) whereas there will be little information on HHV8 and vitamin D. Could you please give your thoughts about this? Page 19: Funding is not correctly spelled.
---

	To conclude, this is a well written and solid protocol that is relevant for the field
--	---

VERSION 1 – AUTHOR RESPONSE

Reviewer: 1

Reviewer Name: Genevieve Mailhot

Institution and Country: University of Montreal, Canada

Please state any competing interests or state 'None declared': None declared

Please leave your comments for the authors below

Please leave your comments for the authors below

This systematic review protocol will bridge a gap in our knowledge on vitamin D and herpes virus infections by providing answers to the following questions: 1) Are deficient or insufficient vitamin D levels associated with the risk of herpes virus infections/reactivation? and; 2) Does vitamin D supplementation protect against herpes virus infections/reactivation? Overall, the protocol is clearly written and the approach looks sound. I only request a few clarifications:

1. Rationale:

The rationale for undertaking this systematic review is thin. Biological plausibility is only briefly touched upon through the general description of the role of vitamin D in innate immune responses. Given that viral infections involve adaptive immune responses as well, I would have expected the authors to summarize the state of knowledge on the role of vitamin D in the adaptive immune system. We have revised the second paragraph of the rationale section on line 61 as follows:

“Recently, some studies have indicated that vitamin D may have potential immunomodulatory effects associated with the regulation of antimicrobial peptides (AMPs).⁶ In previous cell studies, vitamin D induced gene expression of an AMP named Cathelicidin. In response to pathogen exposure, immune cells such as monocytes or macrophages, upregulate vitamin D receptors and enzymes to increase the production of Cathelicidin.⁷⁻⁹ In addition, evidence suggests that vitamin D has some effects on the adaptive immune system. Vitamin D suppresses CD4+ Th 1 lymphocytes and increases Th 2 lymphocytes, and it also intensifies the effect of regulatory T lymphocyte responses.^{10 11}”

The authors cited only two systematic reviews/meta-analyses, which have investigated the role of vitamin D in infections, including a work in individuals with chronic kidney diseases that has addressed infections from a rather broad perspective. In my opinion, the rationale should include a more thorough review of previous observational and interventional studies conducted on vitamin D and viral infections, such as hepatitis C, HIV, and others.

We thank the reviewer for this helpful comment. We have revised the third paragraph of the rationale section on line 74 as follows:

“Vitamin D also shows some anti-infective potential in epidemiological studies. A meta-analysis using original patient data from 25 randomized controlled trials showed that among the general population, vitamin D supplementation reduced the risk of acute respiratory infections.¹⁴ Furthermore, there is some evidence to suggest an anti-infective effect of vitamin D in specific patient groups, such as patients with chronic kidney disease (CKD), HIV or hepatitis C virus (HCV) infection. Among CKD patients receiving dialysis, a case-control study indicated that the risk of herpes zoster reactivation was significantly lower in those who received vitamin D supplementation;¹⁵ another meta-analysis also showed that CKD patients with higher or normal serum vitamin D levels had a lower risk of infection.¹⁶ Among HIV infected patients, lower serum vitamin D levels were also associated with a higher risk of clinical progression to AIDS and all-cause mortality in a cohort study,¹⁷ while vitamin D supplementation didn't affect mortality, CD4 cell count or viral load.¹⁸ For HCV infected patients, serum vitamin D levels were inversely associated with the grade of liver inflammation and the stage of fibrosis,¹⁹ while no protective effect of vitamin D supplementation was seen in a meta-analysis of

clinical trials.²⁰ However, the effect of vitamin D on herpesvirus infection or reactivation in the general population is unclear.”

2. Outcomes:

Is the focus only on the incidence of infection and reactivation or will you consider reporting continuous outcomes such as viral load, clinical symptoms or other measures? This should be made clearer in the protocol. How will you report outcomes, e.g. as risk or odds ratios, as means or mean differences for continuous measures?

We aim to focus on the incidence of infection or reactivation but recognise that these outcome definitions will vary across viruses and studies. Therefore, to clarify the effect measures of the results, we revised the outcomes and prioritization sections on line 199 as follows:

“Regarding the study results, our focus is on the incidence of herpesviruses infection or reactivation. We will report the outcome definition used for each study, and will extract data on the appropriate effect measure. The effect measures will include odds ratios for case-control studies and risk, rate or hazard ratios for cohort studies or clinical trials. If studies report only continuous outcomes such as viral load, then we will summarise these using means or medians as appropriate.”

3. Analysis:

The authors stated that they will analyze the results separately by virus, given their distinct pathogenic mechanisms. If the number of studies is found insufficient to perform such distinct analyses, what will be the alternate strategy?

We plan to summarize existing data in a narrative form for each virus, and describe data by participant characteristics such as immune status. If there are sufficient numbers of studies, and the homogeneity is acceptable, we will consider meta-analyzing the results for each virus, but meta-analysis is not essential.

If the number of included studies is sufficient, please consider including in subgroup analyses the effect of baseline vitamin D status and for intervention trials, study duration.

We agree with this helpful suggestion and have revised the sentence in the data synthesis on line 229 as follows:

“If the number of included studies is sufficient, we will carry out subgroup analyses of subjects with baseline vitamin D status, intervention trials, study durations, different ages, immune statuses, or latitude to explore sources of heterogeneity.”

Reviewer: 2

Reviewer Name: Peter Bergman

Institution and Country: Department of Laboratory Medicine, Karolinska Institutet, Stockholm, SWEDEN

Please state any competing interests or state ‘None declared’: None declared

This protocol describes the plan to perform a systematic review and meta-analysis of the topic of vitamin D and human herpesviruses. The topic is timely, interesting and relevant. Also, the idea to publish the protocol separately is valid. The manuscript is well written and easy to read and understand. However, there are a few minor points that need to be addressed.

We thank the reviewer for these positive comments.

Title: The title reflects the content in a good way.

Abstract: I suggest that the different forms of herpes-virus infections are mentioned in the abstract. Also, it should be mentioned whether placebo-controlled trials will be included or not. This can be important given the inherent problems with bias for all other interventional trials (open, non-controlled, risk of selection bias etc)

We have now listed the nine included human herpesviruses in full in the methods section of the abstract. Regarding placebo-controlled trials, we have now clarified this point in the abstract on line 13 as follows:

“We will use subject headings and keywords to search for synonyms of “vitamin D” and “herpesviruses” (including herpes simplex virus type 1 and 2, varicella-zoster virus, cytomegalovirus, Epstein-Barr virus and human herpesviruses type 6,7 and 8) in Medline, Embase, Global Health, Web of Science, Scopus, and Cochrane Central Register of Controlled Trials, and the grey literature databases Prevention Information & Evidence eLibrary, Open Grey, EThOS, and BASE from inception to 31 August 2019. References to the included articles and relevant systematic reviews will also be examined. Two reviewers will independently screen the study titles and abstracts, and examine the full texts to decide the final eligibility. They will independently extract data from the studies and assess bias using the Cochrane Collaboration approach.”

Strengths and limitations: the concept of grey literature databases needs to be explained somewhere. We have now elaborated on the concept of grey literature databases in the information sources section on line 148:

“To enhance the sensitivity of our search and reduce the risk of publication bias, we will also search grey literature databases such as Open Grey, Prevention Information & Evidence eLibrary, BASE, EThOS and the clinical trials register at ClinicalTrials.gov.”

Rationale: “increased financial burden” needs to be better explained, for whom?

We have now revised the first paragraph of the rationale section on line 52 as follows:

“Reactivation of herpesviruses may induce serious complications. For instance, herpes simplex viruses 1 (HSV-1) can lead to herpetic keratitis, which is the major cause of blindness in developed countries;² Epstein-Barr virus (EBV) may induce nasopharyngeal cancer;³ and Varicella zoster virus (VZV) would cause herpes zoster and post-herpetic neuralgia, which increases financial burdens especially for people older than aged 65 years.⁴ Consequently, investigating immunomodulatory factors associated with infection or reactivation of this virus family is important.”

Reference 6 is not adequate, also it should be written “antimicrobial peptides, AMPs” to follow the standard nomenclature in the field. You could refer to Zasloff, Nature 2002 for an overview of AMPs or similar overviews from a later date. The connection between Vitamin D and AMPs was first described in 2004 by Wang followed in 2005 by Gombart and Weber in three papers published almost at the same time. These papers should be cited.

We have now referenced Zasloff Nature 2002 when describing antimicrobial peptides (AMPs). Also, we have added further information about the role of vitamin D in both innate and adaptive immunity in the rationale section, using the references provided. The reworded text on line 61 is below:

“Recently, some studies have indicated that vitamin D may have potential immunomodulatory effects associated with the regulation of antimicrobial peptides (AMPs).⁶ In previous cell studies, vitamin D induced gene expression of an AMP named Cathelicidin. In response to pathogen exposure, immune cells such as monocytes or macrophages, upregulate vitamin D receptors and enzymes to increase the production of Cathelicidin.⁷⁻⁹ In addition, vitamin D also has some effects on the adaptive immune system. Vitamin D suppresses CD4+ Th 1 lymphocytes and increases Th 2 lymphocytes, and it also intensifies the effect of regulatory T lymphocyte responses.^{10 11}”

Rationale section: please provide some more information why vitamin D would prevent herpes virus reactivation? Are there any mechanistic papers supporting such a claim?

Although the literature is scant, some mechanistic papers suggest that vitamin D can inhibit HSV-1 infection. We now referred to these papers in the rationale section on line 70 as follows:

“Regarding the effects of vitamin D associated AMPs on herpesviruses, a cell study indicated that Cathelicidin decreased HSV-1 viral titres isolated from keratoconjunctivitis patients;¹² furthermore, another cell study also showed that vitamin D supplementation reduced HSV-1 viral load and mRNA expression in HSV-1 infected cells.¹³”

Study design: please comment on the role of placebo-controlled studies or non-controlled studies.

We have modified the sentence in the “Study Design and Characteristics” section on line 109 as follows to clarify this point:

“we will review observational studies, including cohort and case-control studies, and intervention studies with any type of control group, either placebo, active comparator or no treatment, including randomized or non-randomized controlled trials.”

Participants: perhaps a subgroup-analysis should be done for immunocompromised patients?

If there are sufficient numbers of included studies, we will conduct a subgroup analysis by immune status, and now specify this in the Data synthesis and meta-bias(es) section on line 229:

“If the number of included studies is sufficient, we will carry out subgroup analyses of subjects with baseline vitamin D status, intervention trials, study durations, different ages, immune statuses, or latitude to explore sources of heterogeneity.”

Comparators: please comment on placebo or not

For our second question, we will not limit the comparators to placebo only, but will include studies with any control group including active comparators or groups receiving no treatment. This has been clarified in the methods section on line 134 as follows:

“For those receiving oral vitamin D supplementation or treatment, the comparators are those without vitamin D supplementation or treatment or receiving placebo or another active comparator, respectively.”

Outcomes: this is a tricky section since the symptoms of primary infection may vary across the different herpes-viruses. In addition, symptoms of reactivation can also differ between for example HHV6 and HHV8 and HSV-1. Please, comment how you aim to solve this problem in the analysis. My guess is that you will find a majority of studies for a few viruses (HSV1 and 2) whereas there will be little information on HHV8 and vitamin D. Could you please give your thoughts about this?

We agree that the symptoms of infection or reactivation of different HHVs vary. Therefore, we will extract data based on the outcome definitions used in each study by clinical or by laboratory criteria.

Furthermore, we will report results by different virus types and diagnostic methods. We also agree that for some rarely diagnosed herpesviruses infection/reactivation, there may be fewer studies.

Therefore, we have now noted this point in the strengths and limitations section on line 43 as follows:

“The number of sufficient eligible studies may be inadequate, especially for some viruses that are harder to diagnose; also, the included studies may not have an adequate quality of evidence to answer the research questions.”

We will consider this as a limitation in the discussion section of our final systematic review.

Page 19: Funding is not correctly spelled.

We corrected the spelling.

To conclude, this is a well written and solid protocol that is relevant for the field.

Thank you very much.

VERSION 2 – REVIEW

REVIEWER	Genevieve Mailhot University of Montreal, Canada
REVIEW RETURNED	20-Aug-2019

GENERAL COMMENTS	The authors have satisfactorily addressed all of my comments and I have no further concerns. This systematic review, once completed, will make a great contribution to the field.
---

REVIEWER	Peter Bergman Karolinska Institutet, SWEDEN
REVIEW RETURNED	13-Aug-2019

GENERAL COMMENTS	The authors have addressed all of my previous comments in an adequate way.
--